

# An alternative peptone preparation using *Hermetia illucens* (Black soldier fly) hydrolysis: process optimization and performance evaluation

Gaoqiang Liu[1,2,3], Ming Foong Tiang[2,4], Shixia Ma[1,2,3], Zeyan Wei[1,2,3], Xiaolin Liang[1,2,3], Mohd Shaiful Sajab[4,5], Peer Mohamed Abdul[4,5], Xueyan Zhou[1,2], Zhongren Ma[1,2] and Gongtao Ding[1,2]

[1] Key Laboratory of Biotechnology and Bioengineering of State Ethnic Affairs Commission, Biomedical Research Center, Northwest Minzu University, Lanzhou, China
[2] China-Malaysia National Joint Laboratory, Biomedical Research Center, Northwest Minzu University, Lanzhou, China
[3] College of Life Science and Engineering, Northwest Minzu University, Lanzhou, China
[4] Department of Chemical and Process Engineering, Faculty of Engineering and Built Environment, Universiti Kebangsaan Malaysia, Bangi, Selangor, Malaysia
[5] Research Center for Sustainable Process Technology (CESPRO), Faculty of Engineering and Built Environment, Universiti Kebangsaan Malaysia, Bangi, Selangor, Malaysia

Corresponding author
Gongtao Ding, dinggong-tao@outlook.com

## ABSTRACT

**Background.** *Hermetia illucens* (HI), commonly known as the black soldier fly, has been recognized for its prowess in resource utilization and environmental protection because of its ability to transform organic waste into animal feed for livestock, poultry, and aquaculture. However, the potential of the black soldier fly's high protein content for more than cheap feedstock is still largely unexplored.

**Methods.** This study innovatively explores the potential of *H. illucens* larvae (HIL) protein as a peptone substitute for microbial culture media. Four commercial proteases (alkaline protease, trypsin, trypsase, and papain) were explored to hydrolyze the defatted HIL, and the experimental conditions were optimized *via* response surface methodology experimental design. The hydrolysate of the defatted HIL was subsequently vacuum freeze-dried and deployed as a growth medium for three bacterial strains (*Staphylococcus aureus*, *Bacillus subtilis*, and *Escherichia coli*) to determine the growth kinetics between the HIL peptone and commercial peptone.

**Results.** The optimal conditions were 1.70% w/w complex enzyme (alkaline protease: trypsin at 1:1 ratio) at pH 7.0 and 54 °C for a duration of 4 h. Under these conditions, the hydrolysis of defatted HIL yielded 19.25% ±0.49%. A growth kinetic analysis showed no significant difference in growth parameters ($\mu_{max}$, $X_{max}$, and $\lambda$) between the HIL peptone and commercial peptone, demonstrating that the HIL hydrolysate could serve as an effective, low-cost alternative to commercial peptone. This study introduces an innovative approach to HIL protein resource utilization, broadening its application beyond its current use in animal feed.

## INTRODUCTION

*Hermetia illucens* (HI), or black soldier fly (BSF), is increasingly gaining recognition as a remarkable bio-converter and decomposer. This insect is adept at consuming a plethora of organic materials, such as kitchen waste, spoiled fruits and vegetables, animal manure, and food processing waste, and converting them into a rich biomass containing proteins, lipids, amino acids, and peptides (*Meneguz et al., 2018*; *Surendra et al., 2020*; *Amrul et al., 2022*; *Mangindaan, Kaburuan & Meindrawan, 2022*). The dry biomass of *Hermetia illucens* larvae (HIL) boasts crude protein content ranging from 32 to 60% w/w (*Nguyen, Tomberlin & Vanlaerhoven, 2013*; *Al-Qazzaz et al., 2016*; *Ruhnke et al., 2018*). HIL acts as a bulwark against harmful microbes like *Escherichia coli* and *Salmonella*. During the 10–20-day breeding phase, the larvae synthesize a high concentration of antibacterial peptides, creating an inhospitable environment for bacterial and fungal growth (*Secci et al., 2018*; *Cullere et al., 2019*; *Candian, Meneguz & Tedeschi, 2023*; *Candian et al., 2023*). HIL also protects against typical mycotoxins such as deoxynivalenol, fumonisin 1 and 2, and zearalenone in HIL biomass, regardless of the nature of the substrate on which they are reared (*Bosch et al., 2017*; *Camenzuli et al., 2018*; *Leni et al., 2019*). The legislative landscape is also shifting in favor of HIL, with the European Union, the United States, Australia, and Canada having enacted legislation to approve its use in animal feed (*European Union (EU), 2017*; *European Union (EU), 2021*; *Alagappan et al., 2022*). Currently, the breeding industry predominantly uses HIL as a protein-rich alternative to traditional feeds like soybean meal and fishmeal because of its high protein content, feed safety, and cost-effectiveness (*Petrova et al., 2021*). However, only a few studies have been conducted on HIL protein application outside of its use as animal feed.

Commercially available peptones, which are indispensable nitrogen sources in microbial culture media, owe their widespread use to their proven efficiency and stability (*Chapman, Mariano & Macreadie, 2015*; *Fallah, Bahram & Javadian, 2015*). These peptones are typically derived from hydrolyzed sources such as fish, casein, meat, or soybean. However, there are barriers to the use of these sources, including high costs and the potential risk of bovine spongiform encephalopathy (BSE) virus contamination during beef hydrolysis (*Zhang et al., 2023*). These limitations have ignited the pursuit for feasible alternatives.

A growing body of research has begun to explore unconventional sources for peptone derivation. Materials such as wool, tuna heads, and salmon skeletons have been used with different proteases to yield peptones that demonstrate comparable bacterial growth performance to their commercially available counterparts (*Broli et al., 2021*; *Tuysuz et al., 2021*; *Vázquez et al., 2022*). HI are able to grow and mature at a fast growth rate which leads to a high yield of larvae, HIL. It is also found that HIL convert the organic waste efficiently and sustainably into biomass containing high protein content which could be utilized as beneficial nutritional source in peptone-related bacterial cultivation (*Meneguz et al., 2018*). However, despite the significant potential of HIL in nutrient recycling and as a protein-rich feed alternative, very few studies have delved into the preparation of peptones using HIL and hence, this research gap represents an opportunity for exploration, especially considering the impressive credentials of HIL as a sustainable and efficient nutrient source.

In the quest for efficient, cost-effective, and safe solutions to the high costs and biosafety concerns of conventional peptone sources, this study proposes the use of HIL as a promising source for alternative peptone preparation. Recognizing HIL as a prospective alternative peptone source, this study introduces a method to hydrolyze HIL proteins using a variety of protease combinations. The outcome is a protein hydrolysate that has shown potential as a microbial medium as effective as traditional tryptone. This study focuses on optimizing the HIL peptone preparation process, including examining the physicochemical properties of the resultant HIL peptone and conducting a performance evaluation. To assess performance, this study focused on the growth kinetics of *Bacillus subtilis*, *Staphylococcus aureus,* and *Escherichia coli* in HIL-based media.

## MATERIALS & METHODS

### Raw materials and enzymes

The HIL used in this study were purchased from Gansu Guorui Environmental Protection Biotechnology Co. Ltd. (Lanzhou, China). General chemical reagents were obtained from Tianjin Baishi Chemical Industry Co. Ltd. (Tianjin, China) and the biochemical-grade yeast extract and agar peptone (tryptone) were purchased from Beijing Solarbio Science & Technology Co. Ltd. (Beijing, China). Trypsin and papain were purchased from Nanning Donghenghuadao Biotechnology Co. Ltd. (Nanning, China), alkaline protease was purchased from Shandong Longke Bio-Products Co. Ltd. (Linyi, China), and *Bacillus subtilis* ATCC 6051 was purchased from Beijing Bai'ou Bowei Biotechnology Co. Ltd. (Beijing, China). *Staphylococcus aureus* ATCC6583 and *Escherichia coli* ATCC 25922 were preserved by the Biomedical Research Center, Northwest Minzu University (Gansu, China).

### Instruments and equipment

The automatic Kjeldahl nitrogen analyzer (Kjeltec 8200) used in this study was manufactured by FOSS Co. Ltd. in Denmark. Additionally, the Hitachi amino acid analyzer (L-8900) was purchased from Shanghai Baiga Instrument Technology Co. Ltd. and the vacuum freeze dryer (LGJ-20F) employed in the study was purchased from Beijing Songyuan Huaxing Technology Development Co. Ltd.

### Microbial culture medium

The Luria-Bertani (LB) medium containing 10.0 g peptone (tryptone), 5.0 g yeast extract, 10.0 g NaCl, and 15.0 g agar was dissolved in distilled water to a volume of 950 mL. The pH was then adjusted to 7.2–7.4 and more distilled water was added to a final volume of 1000 mL and then sterilized by autoclaving at 120 °C for 20 min.

### Response surface methodology (RSM) experimental design

Data detailing the impact of enzyme complexes (specific enzymes and their proportions), pH, and hydrolysis time on HIL hydrolysis are presented in the File S1. The results of the preliminary factorial experimental test showed the variables of enzyme addition (A) and temperature (B) were the most significant variables on the proteolysis of HIL, so they were selected to be investigated for optimization by response surface methodology. Polynomial

**Table 1  Factors and levels of central composite design.**

| Level | Factors | |
|---|---|---|
| | A: Enzyme addition (%) | B: Temperature (°C) |
| −1.414 | 0.79 | 40.86 |
| −1 | 1.0 | 45 |
| 0 | 1.5 | 55 |
| 1 | 2.0 | 65 |
| 1.414 | 2.21 | 69.14 |

equations, relating the effect of independent variables on maximum hydrolysis (Hm), were obtained after applying the orthogonal least-squares method (Eq. (1)) (*Vázquez et al., 2022*):

$$Y = m_0 + \sum_{i=1}^{n} m_i X_i + \sum_{\substack{i=1 \\ k>i}}^{n-1} \sum_{k=2}^{n} m_{ik} X_i X_k + \sum_{i=1}^{n} m_{ii} X_i^2 \tag{1}$$

where $Y$ is the response evaluated, $m_0$ is the constant coefficient, $m_i$ is the coefficient of linear effect, $m_{ik}$ is the coefficient of combined effect, $m_{ii}$ is the coefficient of quadratic effect, $n$ is the number of variables, and $X_i$ and $X_k$ are the independent variables studied in each case. Five combinations of temperature and enzyme addition were tested, based on previous data provided by the enzyme marketers, to establish the most suitable optimal working conditions. The experimental conditions studied were set at pH 7.0 at a mass ratio of alkaline protease to trypsin at 1:1 under the enzymatic reaction time of 4 h. In all cases, after proteolysis, each liquid hydrolysate was heated at 95 °C for 15 min to inactivate the enzyme. Following the hydrolysis, both the hydrolysate and sediment were subjected to centrifugation. The resultant liquid hydrolysate was then dried using a vacuum freeze dryer (LGJ-20F). This material was distributed evenly on the instrument's rack under a vacuum pressure of 20 Pa. The process involved a linear temperature increase from −45 °C to 25 °C over the course of the 40-hour vacuum freeze-drying cycle. The dried samples were subsequently collected, pulverized, and preserved through vacuum packaging in sterile bags. The Design-Expert 8.0.6 software was used to optimize the hydrolysis process of the HIL. The central composite design experimental setup is shown in Table 1.

### Measurement of main indicators

The physicochemical properties of HIL hydrolysate were obtained by measuring total nitrogen content using the Kjeldahl method, and amino-nitrogen (a-amino) content, which was estimated using the formol titration method according to the methods described by *Kosasih et al. (2018)*. HIL hydrolysate (2 ml) was dissolved in deionized water (five mL) and the pH was adjusted to 8.2 using NaOH (0.1 mol/L). Then, five mL of previously neutralized formaldehyde (pH = 8.2) was added to the mixture and subsequently titrated with NaOH (0.1 mol/L) to attain a pH of 9.2. The total volume of NaOH in the titration was

used to calculate the amino-nitrogen content, as shown in Eq. (1). The protein hydrolysis DH (%) was determined by separately determining the amount of total nitrogen and amino-state nitrogen in the protein solution and calculating the result using Eq. (2):

$$\text{Amino nitrogen (\%)} = \left(\frac{v}{w \times 10}\right) \times M_{(\text{NaOH})} \times 14.008 \qquad (2)$$

where v is the volume of NaOH used in titration, w is the weight of the sample (g), and M(NaOH) indicates NaOH normality. The degree of hydrolysis (DH) was estimated by the ratio of amino nitrogen to total nitrogen in each sample (Eq. (3); *Kosasih et al., 2018*):

$$\text{DH(\%)} = \frac{A(\text{g}/100\text{g})}{T(\text{g}/100\text{g})} \times 100 \qquad (3)$$

where A is amino acid nitrogen in enzymatic digest and T is the total nitrogen in the sample.

### Amino acid analysis
Sample processing involved dissolving the lyophilized powder in pure water, centrifuging it at 1200 rpm for 15 min to remove the protein, and then treating the supernatant with trichloroethylene. After dilution with trichloroethane (TCA), the filtrate was filtered through a 0.22 $\mu$m filter, and the amino acid species and content in the samples were determined through an amino acid analyzer (*National Health Commission of the PRC, China Food and Drug Administration (CFDA), 2016*).

### HIL Peptone and commercial peptone chemical property test
The fat protein content and ash moisture of the samples were determined according to methods 955.04, 934.01, 2003.06 and 942.05 of the AOAC international methods. The ash content of hydrolysates was evaluated using AOAC international method 942.05 (*Cinnif, 1997*). The water solubility index (WSI) was determined using the methods outlined by *Hou (2017)*.

### Effect of HIL peptone and commercial peptone on the growth effect of bacteria
This experiment was conducted in four separate sections, each serving a distinct objective. First, the solubility of peptone was assessed. A precise quantity of 1.0 g of peptone was dissolved in 50 mL of distilled water. The solution was immediately shaken and completely processed within 30 min. In the second section, the clarity and precipitation of the peptone solution were evaluated under alkaline conditions. A solution was prepared by dissolving 2.0 g of peptone in 100 mL of distilled water. The pH was adjusted to 8.0–9.0 using 0.1 mol/L NaOH. Subsequently, the solution was autoclaved at 121 °C for 30 min, followed by cooling to room temperature for observation. The third section involved examining the interaction between peptone and phosphate. A solution was prepared by dissolving 2.0 g of peptone and 0.5 g of $KH_2PO_4$ in 100 mL of distilled water. The pH was adjusted to 7.4–7.6 using 0.1 mol/L NaOH. Similar to the second stage, the solution was autoclaved and cooled before evaluating its clarity and precipitation. In the fourth section, the coagulability of peptone was tested. A 5% aqueous peptone solution was filtered and boiled, and any observed precipitation was noted.

The microorganisms employed in this study are commonly used industrial bacteria (*E. coli* and *B. subtilis*) and pathogenic bacterium (*S. aureus*). These microorganisms were used to assess the efficacy of HIL peptones in comparison to commercial peptones for cultivating microorganisms, taking into account their distinct genera and natures. To determine the ideal bacterial inoculum concentrations, *B. subtilis* (cultivated at 30 °C), *S. aureus* and *E. coli* (cultivated at 37 °C) were separately inoculated into both Luria-Bertani (LB) and HIL peptone liquid media at concentrations ranging from 1.0% to 5.0% (v/v). Following a 12-hour shaking incubation at a consistent agitation speed of 180 rpm, the absorbance was measured at 600 nm to identify the optimal inoculum concentration for each bacterium. Subsequently, *B. subtilis* and *S. aureus* were inoculated at their respective optimal concentrations into LB and HIL peptone liquid media. These cultures were incubated at 30 °C and 37 °C, and samples were collected every two hours for bacterial count determination. The viable bacterial counts (cfu/mL) and corresponding time intervals were then used to construct the growth curves.

Finally, the growth kinetics of the bacterial strains in both media types were evaluated using the Verhulst logistic model (Eq. (4); *Ding et al., 2016*). The growth curves provided the necessary data to determine the maximum biomass concentration ($X_{max}$), maximum specific growth rate ($\mu_{max}$), and lag phase ($\lambda$). These parameters were calculated using OriginPro 2021 to assess the effect of HIL and commercial peptone on bacterial growth:

$$X_t = \frac{X_{max}}{1 + e^{2 + \mu_{max}(\lambda - t)}} \tag{4}$$

where $X_t$ (cfu /mL) is the biomass concentration over time, $X_{max}$ (cfu /mL) is the maximum biomass concentration, $\mu_{max}$ (h$^{-1}$) is the maximum specific growth rate, and $\lambda$ (h) is the lag phase.

### Data analysis software

The experimental results were expressed as mean values $\pm$standard deviations by organizing the raw experimental data in Microsoft Excel and then statistically analyzed using IBM SPSS Statistics 22.0. Response surface optimization was carried out using Design-Expert.V8.0.6.1. Growth curves and fitting curves were plotted using Origin Pro 2021 (9.8.0.200).

## RESULTS AND DISCUSSION

### RSM experimental results

#### Variance analysis and model validation

Using the data in Table 2, the regression equation obtained was, as follows: $Y = 19.56 + 1.52A + 0.74B + 0.56AB - 4.00A^2 - 3.53B^2$.

As shown in Table 3, the regression analysis demonstrated that A, B, A$^2$, and B$^2$ were highly significant, as evidenced by *P*-values less than 0.01. Furthermore, the model itself was found to be highly significant ($P < 0.01$). The lack-of-fit term was not significant ($P > 0.05$), indicating that the model provided a good fit to the experimental data and had good stability. The determination coefficient ($R^2$) was 0.9752, suggesting that 97.52% of the variability in the response could be explained by the model. The adjusted $R^2$ was 0.9574, further confirming the model's reliability in predicting experimental outcomes. The model

**Table 2 Central composite design test results.**

| Serial number | Factors | | Hydrolysis predictions (%) | Hydrolysis degree actual value (%) |
|---|---|---|---|---|
| | A (%) | B (°C) | | |
| 1 | 0 | 0 | 19.56 | $19.42 \pm 0.27$ |
| 2 | 1.414 | 0 | 13.71 | $14.13 \pm 0.31$ |
| 3 | 0 | 1.414 | 20.61 | $19.07 \pm 0.27$ |
| 4 | −1 | −1 | 13.86 | $14.97 \pm 0.27$ |
| 5 | 0 | 0 | 19.56 | $18.97 \pm 0.31$ |
| 6 | −1 | 1 | 14.22 | $15.20 \pm 0.41$ |
| 7 | 1 | −1 | 15.78 | $17.33 \pm 0.27$ |
| 8 | 1 | 1 | 18.38 | $20.40 \pm 0.27$ |
| 9 | 0 | 0 | 19.56 | $18.62 \pm 0.15$ |
| 10 | 0 | 0 | 19.56 | $18.91 \pm 0.13$ |
| 11 | 0 | −1.414 | 18.51 | $17.33 \pm 0.15$ |
| 12 | −1.414 | 0 | 9.41 | $10.63 \pm 0.15$ |
| 13 | 0 | 0 | 19.56 | $18.20 \pm 0.27$ |

**Table 3 Variance analysis results of response surface quadratic regression equation model.**

| Source | Sum of squares | Degree of freedom | Mean square | F value | P-value |
|---|---|---|---|---|---|
| Models | 46.77 | 5 | 9.35 | 54.99 | <0.0001[*] |
| A-Enzyme addition | 4.13 | 1 | 4.13 | 24.25 | 0.0017[*] |
| B-Temperature | 2.19 | 1 | 2.19 | 12.90 | 0.0088[*] |
| AB | 0.20 | 1 | 0.20 | 1.17 | 0.3159 |
| $A^2$ | 7.04 | 1 | 7.04 | 41.41 | 0.0004[*] |
| $B^2$ | 35.40 | 1 | 35.40 | 208.07 | <0.0001[*] |
| Residuals | 1.19 | 7 | 0.17 | | |
| Misfit term | 0.97 | 3 | 0.32 | 6.02 | 0.0578 |
| Pure error | 0.22 | 4 | 0.05 | | |
| Total difference | 47.96 | 12 | $R^2 = 0.9752; R^2_{adj} = 0.9574$ | | |

**Notes.**
 *Extremely significant difference ($P < 0.01$).

effectively fit the real response surface, thereby elucidating the interrelationship among enzyme concentration, temperature, and hydrolysis. These results showed the model was suitable for analyzing and forecasting the results of the defatted HIL hydrolysis process.

### *Response surface analysis*
This study revealed a notable trend: the hydrolysis degree, at a constant enzyme concentration, initially increased and later decreased as the temperature varied between 45 °C and 60 °C (Fig. 1). As shown in Fig. 1 by the black point in the figure, the hydrolysis degree peaked at 55 °C. The underlying mechanism of this trend is the progressive depletion of the substrate concentration, as it was hydrolyzed by the increasing enzyme quantity until it was fully hydrolyzed. The concentration of free amino nitrogen and the hydrolysis degree
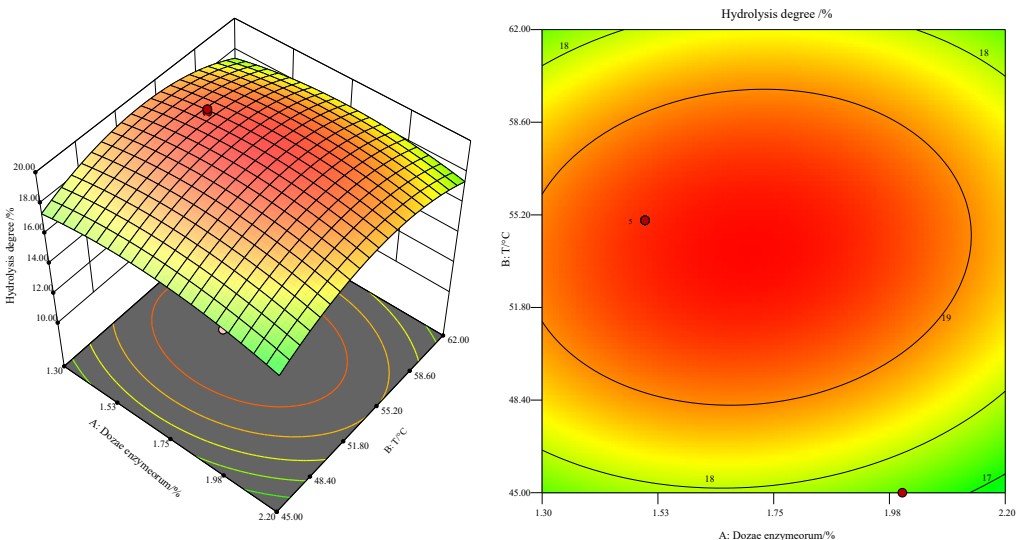

**Figure 1** Response surface and contours of the effect of enzyme addition and temperature on the degree of hydrolysis.

reached a plateau. Moreover, the higher slope for temperature in comparison to enzyme dosage in the response surface analysis indicated a more pronounced impact of temperature on the enzyme-catalyzed reaction rate.

### *Optimization and validation*

The predicted optimal conditions using the regression equation for hydrolysis were determined to be an enzyme dosage of 1.7% (w/w) at 54 °C, with a theoretical hydrolysis degree of 19.75% ±0.995%. Three validation experiments under these conditions yielded a hydrolysis degree of 19.25% ±0.49% for the defatted HIL. The relative error between the observed and theoretical values was 2.60%, and the difference was statistically non-significant ($P > 0.05$). These results provide compelling evidence of the model's practicality and reliability for experimental optimization and for identifying the best hydrolysis conditions.

## General characterization of HIL peptone and commercial peptone

As delineated in Table 4, the HIL peptone exhibited a darker color compared to commercial peptone. This variation is likely attributed to the higher sugar content in the raw material, instigating Maillard reactions and caramelization under elevated temperature conditions. These reactions intensify the color of the final product. Nevertheless, the product's appearance aligns with the quality standard of peptone and does not adversely influence the growth or reproduction of microorganisms. Optimal peptone coloration should steer clear of extreme darkness, as it could interfere with morphological observation and the growth and reproduction assays of microorganisms. In assessments of clarity, sedimentation, and solubility, both HIL peptone and commercial peptone demonstrated comparable

**Table 4   Test results of general indexes of HIL peptone and tryptone.**

| Item | HIL peptone | Tryptone |
|---|---|---|
| Color | Light brown | Light yellow |
| Clarity | Clarity and transparency | Clarity and transparency |
| Precipitation | Small amount of precipitation | No precipitation |
| Dissolution time | <30 min | <30 min |
| pH | 8.83 ± 0.03 | 7.10 ± 0.01 |
| Alkaline precipitation | A little precipitation | No precipitation |
| Phosphate precipitation | Little precipitation | No precipitation |
| Coagulable peptone | No precipitation | No precipitation |

results, displaying clarity, transparency, and full dissolution within 30 min. Clarity and transparency are fundamental properties for peptones used as biochemical reagents in microbial culture so microbial morphologies can be observed during cultivation. A clouded peptone solution may compromise the accuracy of observational results (*China Committee for Standardization of Biological Products, 2000*).

The pH value of HIL peptone was 8.8, while the pH value of commercial peptone was 7.1. *Bruno et al. (2019)* indicated that the pH of the contents varies based on the position of intestinal lumen, at which the anterior and central portion of the midgut is acidic, whereas it is alkaline in the posterior portion of the midgut. Hence, the metabolites excreted by the worms during growth were alkaline. In addition, the pH of HIL peptone prepared was not adjusted, likely contributing to the weakly alkaline appearance of the HIL peptone. The HIL protein hydrolysate contains a diverse range of digestive enzymes including cellulases, lipases, $\alpha$-amylases, proteases, and pancreatic proteases. In the process of preparing HIL peptone, the pH value of the reaction system was not adjusted, leading to alkaline precipitation. Both alkaline and phosphate precipitates were observed in the HIL peptone, while pancreatic protein hydrolysate did not exhibit such precipitation. This phenomenon may be attributed to the presence of a considerable amount of chitin, which cannot be removed, in the larvae stage of HIL, resulting in the formation of slight precipitates in the protein hydrolysate solution. Nevertheless, both HIL peptone and commercial peptone met the growth requirements of microorganisms in terms of coagulable peptones.

## Chemical properties of HIL peptone and commercial peptone

Table 5 indicates the subtle differences between HIL peptone and commercial peptone composition, including their respective moisture, ash, total nitrogen, amino nitrogen, phosphorus, and chloride profiles. These elemental constituents within HIL peptone aligned with the quality benchmarks for peptone (*Chen, 1995*). The moisture content of HIL peptone and commercial peptone were recorded at 4.34% and 4.89%, respectively. Elevated moisture levels in peptone increased the occurrence of clumping phenomena, resulting in the degradation and eventual spoilage of its constituent elements, negatively impacting microbial growth and product synthesis. These results indicate moisture is an essential control parameter. The ash content of HIL peptone was 12.77%, which was similar to the 13.08% ash content of commercial peptone.

**Table 5  Chemical properties of HIL peptone and tryptone.**

| Item | HIL peptone | Tryptone | Peptone for biochemical reagent |
|---|---|---|---|
| Moisture (%) | 4.34 ± 0.01[b] | 4.89 ± 0.02[a] | <5.0 |
| Ash content (%) | 12.77 ± 0.23[b] | 13.08 ± 0.08[a] | <15.0 |
| Total nitrogen (%) | 12.38 ± 0.12[b] | 12.70 ± 0.12[a] | >12.0 |
| Amino nitrogen (%) | 2.53 ± 0.02[b] | 3.70 ± 0.03[a] | >2.5 |
| Phosphorus content (%) | 0.53 ± 0.03[b] | 0.75 ± 0.03[a] | – |
| Chloride content (as chlorine) (%) | 1.43 ± 0.03[a] | 0.40 ± 0.12[b] | ≤2.0 |

Notes.

Data for HIL peptone and tryptone were derived from this study, while data for biochemical reagent peptone were sourced from literature (*Chen, 1995*). Different lowercase letters in the shoulder labels of data in the same row in the table indicate significant differences ($P < 0.05$), and the same letters indicate no significant differences ($P > 0.05$).

The total nitrogen content of HIL peptone and commercial peptone were 11.38% and 12.70%, respectively. Peptone plays a crucial role as the primary nitrogen source for nutrients during microbial growth and metabolism, so nitrogen content is a vital quality assessment criterion (*Taskin et al., 2016*). Phosphorus content in both HIL peptone and commercial peptone fell below 1.0%. Phosphorus is a crucial nutrient for the synthesis of microbial biomolecules, such as nucleic acids and proteins, and serves as a crucial pH buffer substance in the culture medium, highlighting the importance of controlling phosphorus levels for microbial growth and metabolism. The chloride content of HIL peptone and commercial peptone were 1.43% and 0.40%, respectively, which both comply with the microbial culture medium standards of <2.0 (*Chen, 1995*). It is noteworthy that the differences in microbial growth performances using HIL peptone and commercial peptone were insignificant even though the chloride content of HIL peptone was relatively higher than the commercial peptone.

## Water solubility index of HIL peptone and commercial peptone

Figure 2 shows the water solubility index (WSI) of HIL peptone and commercial peptone at various pH levels. Both HIL peptone and commercial peptone exhibited an initial increase in water solubility, followed by a subsequent decrease. At pH 9.0, the HIL protein hydrolysate reached its highest WSI at 5.20%, while commercial peptone reached a peak value of 6.46% at pH 7.0. These comparable values and trends suggest functional parallels between the hydrolysates. The lowest solubility for both hydrolysates was observed at pH 1.0. At this pH value, the absence of repulsive electrostatic forces between protein molecules leads to their aggregation or precipitation, indicating the isoelectric point of the proteins. This phenomenon implies that under non-extreme pH conditions, the properties of HIL peptones remain stable.

## Analysis of amino acids in the HIL hydrolysate

Figure 3 delineates the respective amino acid compositions of HIL peptone, commercial peptone, and bovine blood peptone. The HIL peptone has a significantly higher percentage of tyrosine (12.92%), arginine (11.59%), leucine (11.23%), and phenylalanine (9.38%) compared to the other two peptones. Conversely, the commercial peptone has significantly

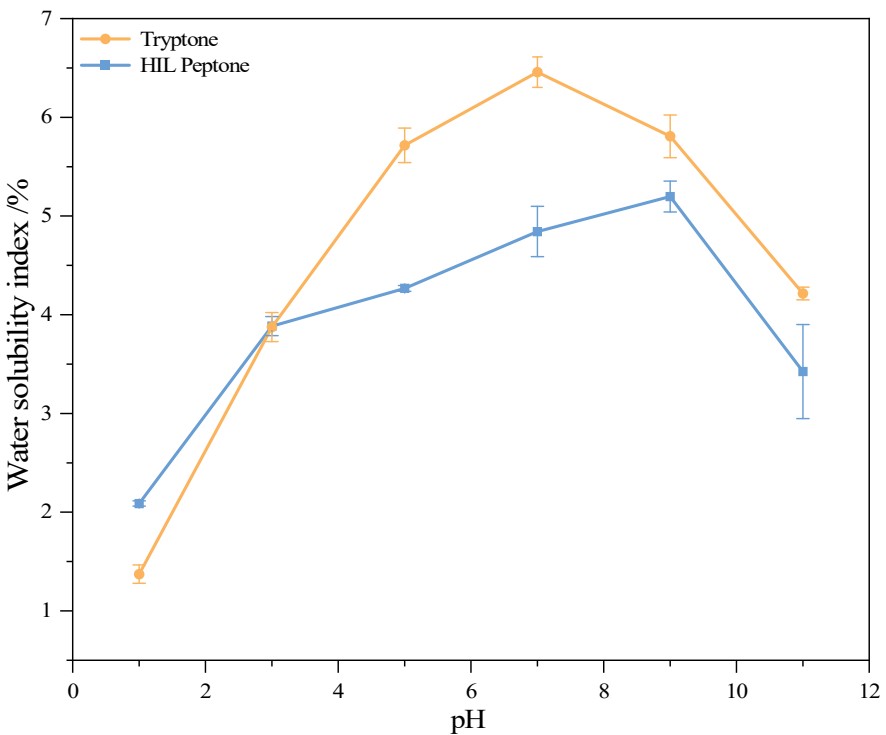

**Figure 2** **Water solubility index of two peptones at different pH values.**

higher levels of glutamic acid (17.53%), glycine (16.79%), aspartic acid (12.22%), and alanine (10.10%) than the other two peptones. The bovine blood peptone, as analyzed by *Rezaee et al. (2023)* primarily contains lysine (18.54%), glutamic acid (12.83%), aspartic acid (10.95%), and alanine (8.706%). The amino acids present in these peptones are integral to microbial growth and perform critical roles such as aiding microbial protein synthesis, catalyzing enzymatic reactions, and providing antioxidative properties. The work of *Triani et al. (2021)* indicates that enzymatic hydrolysis can effectively elevate amino acid content in BSF prepupae. *Triani et al. (2021)* also found that protein hydrolysate is not only abundant in amino acids, but also contains short chain peptides, which are more readily assimilable by microorganisms.

As shown in Fig. 4, a principal component analysis (PCA) revealed amino acid differences between HIL peptone, bovine blood peptone, and commercial peptone based on 22 different amino acids. The direction and length of the arrows in the figure indicate the direction of the principal components and the contribution of each of the differences. The HIL peptone exhibited distinct characteristics from the bovine blood peptone and commercial peptones, implying a low correlation between them. The primary differences among the three peptones are attributable to the content of tyrosine, arginine, and leucine. Bovine blood and commercial peptones shared a high correlation, as indicated by the close proximity of their respective arrows, with glycine, glutamic acid, and lysine highlighted as the principal amino acids responsible for their dissimilarity. Despite substantial differences

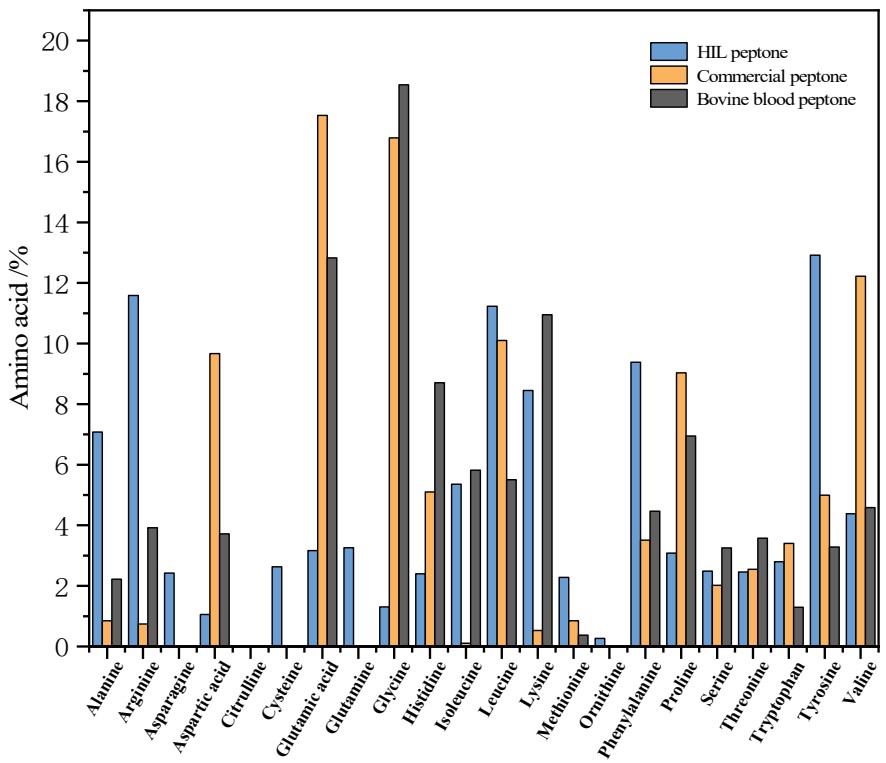

**Figure 3** Percentage contribution of each amino acid to the total amino acids in the peptone sample (HIL peptone, commercial peptone, and bovine blood peptone).

in amino acid composition, the HIL peptone and commercial peptone did not significantly impact microbial growth. This fact enhances the appeal of HIL peptone as an economically viable and environmentally sustainable substitute to conventional peptones. HIL peptone is particularly beneficial for microbial cultures that demand swift nutrient absorption and energy conservation.

In summary, the amino acid composition variations among the peptones could exert unique effects on the growth and metabolism of microorganisms. This is particularly relevant in the field of industrial biotechnology, where modifying the nutrient composition of the growth medium could optimize specific microbial processes. Future research could exploit this potential by investigating the impacts of these peptones on a range of industrially relevant microorganisms.

### Effect of HIL peptone and commercial peptone on the growth effect of the strain

Figure 5 and the File S1 show that the optimum inoculum volume for the three species of bacteria were 2.0% (v/v) in HIL peptone medium and 3.0% (v/v) in commercial peptone. The seed cultures of *B. subtilis*, *S. aureus,* and *E. coli* were inoculated into HIL peptone liquid medium and LB liquid medium at their respective optimum inoculation levels. Growth curves were measured every two hours, revealing analogous growth trends in HIL

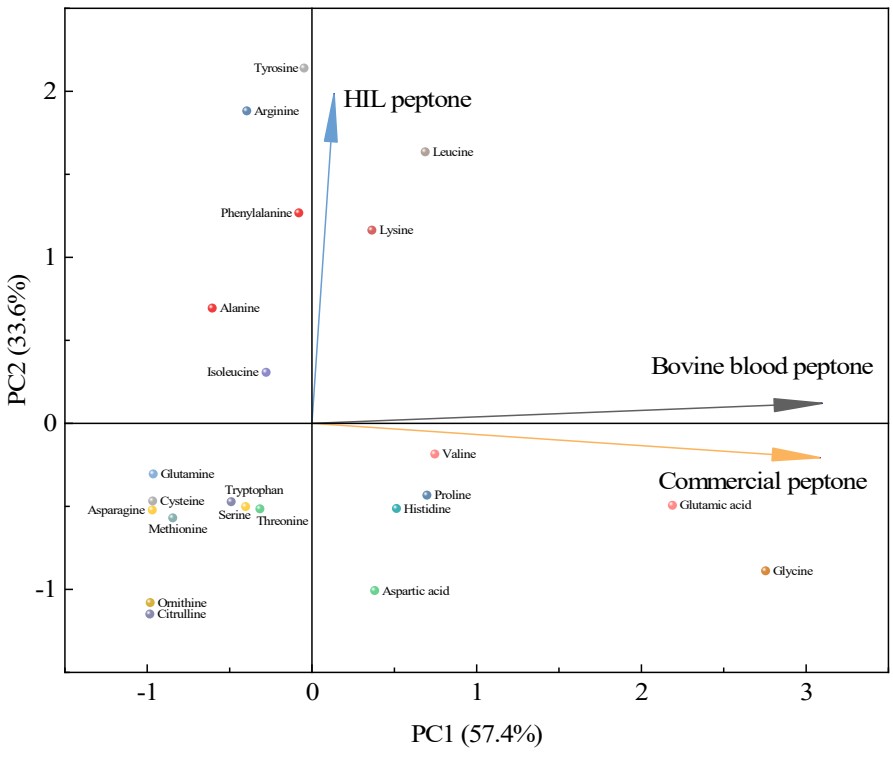

**Figure 4  Principal component analysis for assessing similarities among the HIL peptone, commercial peptone, and bovine blood peptone based on amino acid compositions.**

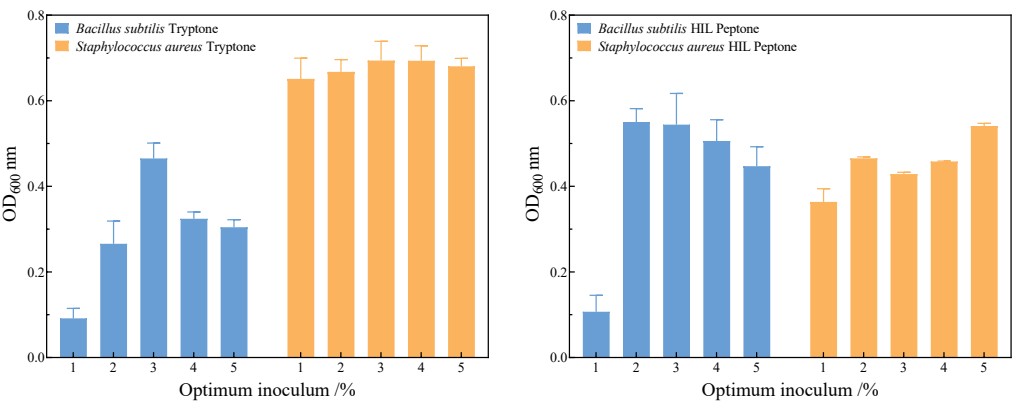

**Figure 5  Determination of the optimum inoculum volume of *Bacillus subtilis* and *Staphylococcus aureus* in tryptone (A) and HIL peptone (B).**

peptone and commercial peptone for these bacteria, as shown in Fig. 5. The growth phases included a lag phase, logarithmic growth phase, and stationary phase.

The use of peptone in media tends to enhance growth, with peptone's key role being to serve as an organic nitrogen source that fulfills bacterial cellular requirements for amino

**Table 6 Growth kinetic parameters generated from the logistic model.**

| Bacterial strain | Medium | $\mu_{max}$ (h$^{-1}$) | $X_{max}$ (cfu/mL) | $\lambda$ (h) | $R_{adj^2}$ |
|---|---|---|---|---|---|
| *Bacillus subtilis* | Tryptone | 0.35 ± 0.04[a] | 8.37 ± 0.32[a] | 15.28 ± 0.54[a] | 0.9880 |
| | HIL peptone | 0.42 ± 0.05[a] | 7.98 ± 0.22[b] | 14.67 ± 0.50[a] | 0.9888 |
| *Staphylococcus aureus* | Tryptone | 0.54 ± 0.04[a] | 4.13 ± 0.07[a] | 4.48 ± 0.24[a] | 0.9967 |
| | HIL peptone | 0.47 ± 0.03[a] | 4.29 ± 0.08[a] | 3.97 ± 0.26[a] | 0.9964 |
| *Escherichia coli* | Tryptone | 0.73 ± 0.03[a] | 6.24 ± 0.04[a] | 2.59 ± 0.14[a] | 0.9980 |
| | HIL peptone | 1.11 ± 0.09[b] | 5.99 ± 0.05[a] | 3.57 ± 0.16[b] | 0.9967 |

**Notes.**

In the table, different letters on the shoulder labels of the same strain and the same column data indicate significant differences ($P < 0.05$), and the same letters indicate no significant differences ($P > 0.05$).

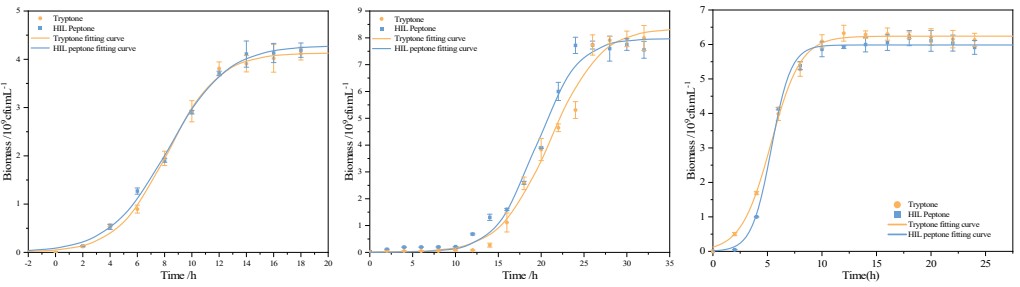

**Figure 6 Growth curve and fitting curve of (A)** *S. aureus,* **(B)** *B. subtilis,* **and (C)** *E. coli.*

acids and peptide. This study performed a comparative analysis of the logistic kinetic parameters for bacterial strains grown in both commercial peptone and HIL peptone mediums. This comparison, outlined in Table 6, included the maximum population size ($X_{max}$), maximum specific growth rate ($\mu_{max}$), and lag phase ($\lambda$) of *S. aureus, B. subtilis* and *E. coli* cultivated in both mediums. An intriguing finding was the higher biomass yield ($X_{max} = 4.29 \times 10^9$ cfu) of *S. aureus* cultured in HIL peptone compared to that in commercial peptone ($X_{max} = 4.13 \times 10^9$ cfu). While other growth parameters did not show significant differences ($p > 0.05$), the maximum specific growth rate of *B. subtilis* in HIL peptone ($\mu_{max} = 0.42$ h$^{-1}$) was slightly higher than that in commercial peptone ($\mu_{max} = 0.32$ h$^{-1}$). The cultivation of *E. coli* with HIL peptone also showed enhanced $\mu_{max}$ ($\mu_{max} = 1.11$ h$^{-1}$ in HIL peptone and $\mu_{max} = 0.73$ h$^{-1}$ in commercial peptone), suggesting potential advantages of HIL peptone for fast-growing cultures. The growth curves in Fig. 6 illustrate the variation in growth patterns of different bacterial strains under different culture conditions. The fit of the logistic model with the growth data of all bacterial strains examined substantiated its suitability for this study. Furthermore, a shorter lag phase was observed for both *S. aureus* ($\lambda = 3.97$ h) and *B. subtilis* ($\lambda = 14.67$ h) using HIL peptone compared with the commercial peptone medium. This expedited

lag phase and the accelerated increase in bacterial concentration during the logarithmic growth phase underscore the potential of HIL peptone for microbial cultivation.

Based on the fact that bacteria cultured in HIL peptone yielded similar results to those cultured in commercial peptone in terms of growth profile and maximum bacterial growth, HIL peptone could serve as an effective substitute for meat-derived commercial peptone. This conclusion aligns with the proposition of *Vázquez et al. (2020)* that aquaculture peptones could be used as alternatives to meat-derived commercial peptone. In similar studies, *Vázquez et al. (2020)* successfully used thermal or enzymatic techniques to derive peptone from fish byproducts for the cultivation of *Lactobacillus* bacteria. *Rezaee et al. (2023)* also demonstrated that bovine blood protein hydrolysate, obtained *via* integrated heat and enzymatic treatment, showed a marked physico-chemical similarity to commercial peptone, a meat-derived product.

Fish protein products are often marred by rancidity and the unpleasant odor of fish oil, as pointed out by *Bridson & Brecker (1970)*. HIL peptone may present a superior option, although further research should be conducted on a broader range of bacteria and their metabolites in fermentation for a more comprehensive assessment of the potential of HIL peptone as an alternative to meat-derived commercial peptone.

## CONCLUSIONS

A recent market analysis revealed that dried HIL is globally used as very low-value animal feed or fishmeal. This study provides another potential HIL application, highlighting the value-added potential of HIL peptone in both scientific and industrial applications. The results of this study indicate that HIL peptone could be applied as a high-value medium as the cultivation of *B. subtilis*, *S. aureus*, and *E. coli* with HIL peptone achieved similar microbial growth performance as those using commercial peptone. The extracted HIL peptone could thus be used as a cultivation medium as an alternative to conventional commercial peptone media.

## ACKNOWLEDGEMENTS

During the preparation of this work the authors used Open AI-ChatGPT 4.0 in order to improve the manuscript language. After using this tool, the authors reviewed and edited the content as needed and take full responsibility for the content of the publication.

### Funding

This work was supported by the Innovation Team Grant of Biopharmaceutical and Material Engineering from Northwest Minzu University and the Bioengineering First-class Disciplines Grant of Northwest Minzu University (81080334). The funders had no role in study design, data collection and analysis, decision to publish, or preparation of the manuscript.

## Grant Disclosures

The following grant information was disclosed by the authors:

Innovation Team Grant of Biopharmaceutical and Material Engineering from Northwest Minzu University and the Bioengineering First-class Disciplines Grant of Northwest Minzu University: 81080334.

## Competing Interests

The authors declare that they have no known competing financial interests or personal relationships that could have appeared to influence the work reported in this paper.

## Author Contributions

- Gaoqiang Liu conceived and designed the experiments, performed the experiments, analyzed the data, prepared figures and/or tables, authored or reviewed drafts of the article, and approved the final draft.
- Ming Foong Tiang conceived and designed the experiments, performed the experiments, analyzed the data, prepared figures and/or tables, and approved the final draft.
- Shixia Ma conceived and designed the experiments, performed the experiments, analyzed the data, prepared figures and/or tables, authored or reviewed drafts of the article, and approved the final draft.
- Zeyan Wei performed the experiments, analyzed the data, prepared figures and/or tables, and approved the final draft.
- Xiaolin Liang analyzed the data, prepared figures and/or tables, and approved the final draft.
- Mohd Shaiful Sajab analyzed the data, authored or reviewed drafts of the article, and approved the final draft.
- Peer Mohamed Abdul analyzed the data, authored or reviewed drafts of the article, and approved the final draft.
- Xueyan Zhou conceived and designed the experiments, authored or reviewed drafts of the article, and approved the final draft.
- Zhongren Ma analyzed the data, authored or reviewed drafts of the article, and approved the final draft.
- Gongtao Ding conceived and designed the experiments, analyzed the data, authored or reviewed drafts of the article, and approved the final draft.

## Data Availability

The raw data are available in the Supplemental File.

## Supplemental Information

Supplemental information for this article can be found online at http://dx.doi.org/10.7717/peerj.16995#supplemental-information.

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
