# Peer review of "An alternative peptone preparation using Hermetia illucens (Black soldier fly) hydrolysis: process optimization and performance evaluation"

_PeerJ, doi:10.7717/peerj.16995_

## Round 0.1 · original submission · Major Revisions

Please make corrections and submit the manuscript with the changes marked.

·

Basic reporting

The study is well-designed and well done. It provides a novel avenue for utilizing insect materials.

Experimental design

The experimental design is well done. The only comment I have is that it would have been good if the authors had tested the efficacy of HIL peptones on a wider range of microorganisms, particularly lactic acid bacteria (LAB) whose growth results in the production of acids. It would be interesting to know what the effect on HIL peptone would be during the generation of organic acids in the growth medium of LAB.

Validity of the findings

All underlying data have been provided; they are robust, statistically sound, & controlled. I have no negative comments on the validity of the findings.

Additional comments

I have included some minor editorial comments on the attached manuscript file for the authors attention.

Reviewer 2 ·

Basic reporting

I think the paper do not have fails. It is important to change some part to allow the reader to understand better the methods and results.

Experimental design

The experimental design is fine and it makes sense with the innovative application of Hermetia meal as starter for produce peptone.

Validity of the findings

The findings are innovative and clear.

Additional comments

Comment for Authors


Back ground: I will change “seafood” with “aquaculture”

Line 27: I will change “seafood” with “aquaculture”

Line 48-49 maybe there is a mistake with the space between “rich b ….iomass”
Line 49-50( add more references for this part) (I can suggest some example

Nguyen, T. T., Tomberlin, J. K., & Vanlaerhoven, S. (2013). Influence of resources on Hermetia illucens (Diptera: Stratiomyidae) larval development. Journal of Medical Entomology, 50(4), 898-906.

Meneguz, M., Schiavone, A., Gai, F., Dama, A., Lussiana, C., Renna, M., & Gasco, L. (2018). Effect of rearing substrate on growth performance, waste reduction efficiency and chemical composition of black soldier fly (Hermetia illucens) larvae. Journal of the Science of Food and Agriculture, 98(15), 5776-5784.

Surendra, K. C., Tomberlin, J. K., van Huis, A., Cammack, J. A., Heckmann, L. H. L., & Khanal, S. K. (2020). Rethinking organic wastes bioconversion: Evaluating the potential of the black soldier fly (Hermetia illucens (L.))(Diptera: Stratiomyidae)(BSF). Waste Management, 117, 58-80.


Line 54: I will suggest for antimicrobial peptide two recent papers:

Candian, V., Savio, C., Meneguz, M., Gasco, L., & Tedeschi, R. (2023). Effect of the rearing diet on gene expression of antimicrobial peptides in Hermetia illucens (Diptera: Stratiomyidae). Insect Science.

Candian, V., Meneguz, M., & Tedeschi, R. (2023). Immune responses of the black soldier fly Hermetia illucens (L.)(Diptera: Stratiomyidae) reared on catering waste. Life, 13(1), 213.


Line 56: I will suggets more papers for mycotoxin:

Camenzuli, L., Van Dam, R., De Rijk, T., Andriessen, R., Van Schelt, J., & der Fels-Klerx, V. (2018). Tolerance and excretion of the mycotoxins aflatoxin B1, zearalenone, deoxynivalenol, and ochratoxin A by Alphitobius diaperinus and Hermetia illucens from contaminated substrates. Toxins, 10(2), 91.

Bosch, G., Van Der Fels-Klerx, H. J., de Rijk, T. C., & Oonincx, D. G. (2017). Aflatoxin B1 tolerance and accumulation in black soldier fly larvae (Hermetia illucens) and yellow mealworms (Tenebrio molitor). Toxins, 9(6), 185.


Line 58: For European legislation:
Authorization for aquaculture:
Commission Regulation (EU) 2017/893 of 24 May 2017 amending Annexes I and IV to Regulation (EC) No 999/2001 of the European Parliament and of the Council and Annexes X, XIV and XV to Commission Regulation (EU) No 142/2011 as regards the provisions on processed animal protein (Text with EEA relevance. )
https://eur-lex.europa.eu/legal-content/EN/TXT/?uri=CELEX:32017R0893

Authorization for pigs and poultries:
Commission Regulation (EU) 2021/1372 of 17 August 2021 amending Annex IV to Regulation (EC) No 999/2001 of the European Parliament and of the Council as regards the prohibition to feed non-ruminant farmed animals, other than fur animals, with protein derived from animals (Text with EEA relevance)
https://eur-lex.europa.eu/legal-content/EN/TXT/?uri=CELEX%3A32021R1372

line 167: Bacillus subtilis in italics, also in line 171 and line 172

Line 195: Are you meaning A= m0 coefficient? And B= mjk of combined effect in the regression equation as indicated in the line 114?
Line 195 to line 203 please refer to Table 3, I got this point by myself but it is important to ciet the table un the text.
Line 207: “as showed in Figure 1 by the black point in the figure.
Line 233 to 234: I think it is due to the faeces inside of the larvae, the larvae usually produce nitric acid as faeces. It is more related to it than feeding requirement. Moreover, the posterior part of the gut of larvae is basic pH, to see :

Bruno, D., Bonelli, M., De Filippis, F., Di Lelio, I., Tettamanti, G., Casartelli, M., ... & Caccia, S. (2019). The intestinal microbiota of Hermetia i llucens larvae is affected by diet and shows a diverse composition in the different midgut regions. Applied and environmental microbiology, 85(2), e01864-18.


Line 245-246: do you think chloride content of 1.43 can influence the quality of HIL peptone compared to 0.40 of tryptone? Please explain it

Line 257: Instead of this graph is better to indicate the amino acid composition in a table with values. Please change it.

Line 265 to line 268: it is normal beaciuse of if you read paper about amino acid of HI you will see the composition of HI meal is completely different. Please find references about amino ac id composition of the meal of HI and comparison with other animal meal.

Annotated reviews are not available for download in order to protect the identity of reviewers who chose to remain anonymous.

Reviewer 3 ·

Basic reporting

The paper it's clear, well written in scientific English.
The figures and table provide useful information to the paper.
The reference to the literature could be improved in some points.

Figure: the explanation of the figure could be more clear, example indicating the meaning of HIL (figure 4 and etc) and indicating the statistical representation of the data.
Tables: reference in the text to table 3 and 4 is not provided.

Experimental design

The novelty of the study it’s clear and strong.

Material and methods, Line 110-112: provide more detail on the methodology. Has been used a software? In case add details please. Is the same mentioned at line 127?
Material and methods, Line 132: reference incorrect, change with Kosasih et al., (2018)
Material and methods, Line 138: change in equation 2 or Eq. 2.
Material and methods, Line 180: change in Equation 4 or Eq.4
Material and methods, Line 188-189: specify what statistical elaborations has been applied

Validity of the findings

The findings are interesting and promising.
Brief analysis on cost evaluation of the innovative solution proposed should be provided, comparing them with the cost of the material now used.
Why for this and not for animal feed?

Additional comments

Here some suggestions

Insert a space between the last word and the parenthesis containing the references.

Abstract, Line 28: better explain the concept of “value-added applications”

Introduction, line 46-47: provide reference concerning the different rearing material already under study on the literature
Introduction, line 51-52: “HIL acts as a bulwark against harmful microbials like Escherichia coli and Salmonella” provide reference

Results and discussion, Line 201: remove space
Results and discussion, Line 227: meaning of the number 25?
Results and discussion, Line 229-231: it is possible to provide a reference?
Results and discussion, Line 243: elaborate more on this paragraph (3.3)
Results and discussion, Line 262: the reference is incomplete (Nasim Rezaee et al.,)
Results and discussion, Line 262: follow the guideline for refeences
“For three or fewer authors, list all author names (e.g. Smith, Jones & Johnson, 2004). For four or more, abbreviate with ‘first author’ et al. (e.g. Smith et al., 2005).” Etc…

Results and discussion, Line 273: typo (amongst)
Results and discussion, Line 312: the reference is incomplete (Vazquez et al.,)
Results and discussion, Line 313-314: typos
Results and discussion, Line 315: the reference is incomplete
Results and discussion, Line 319: follow the guideline for references

Annotated reviews are not available for download in order to protect the identity of reviewers who chose to remain anonymous.

---

## Round 0.2 · Minor Revisions

Research question clear and interesting.

About the line 76-77 of the introduction, i would suggest to explain more why HIL is presented as "sustainable and efficient nutrient source.

·

Basic reporting

I am happy with the rebuttal responses provided by the authors.

Experimental design

I am happy with the rebuttal responses provided by the authors.

Validity of the findings

I am happy with the rebuttal responses provided by the authors.

Additional comments

I am happy with the rebuttal responses provided by the authors.

Reviewer 2 ·

Basic reporting

The authors reply to all my requests.
They complete the references.

Experimental design

Well done

Validity of the findings

The findings are interesting for the future application of BSF meal as a raw material for peptone production.

Reviewer 3 ·

Basic reporting

The paper has been improved,
the information are clear and well-written in english.
I appreciate the explanation of table and figures, really helps to reader to a easy interpretation.

Experimental design

Research question clear and interesting.
About the line 76-77 of the introduction, i would suggest to explain more why HIL is presented as "sustainable and efficient nutrient source".

Validity of the findings

Richness of data, high-quality images.

---

## Round 0.3 · accepted · Accept

The authors made the suggested changes and therefore we approved the manuscript in its current state.